# Antibody responses to Sinopharm/BBIBP-CorV in pregnant mothers in Sri Lanka

Chandima Jeewandara[1], K. A. Chintha S. Jayampathi[2], Thushali Ranasinghe[1], Inoka Sepali Aberathna[1], Banuri Gunasekara[1], Saubhagya Danasekara[1], Thashmi Nimasha[1], Heshan Kuruppu[1], Osanda Dissanayake[1], Nayanathara Gamalath[1], Dinithi Ekanayake[1], Jewantha Jayamali[1], Gayasha Somathilake[1], Dinuka Guruge[2], Ruwan Wijayamuni[2], Achala Kamaladasa[1], Graham S. Ogg[3], Gathsaurie Neelika Malavige[1,3]*

1 Allergy Immunology and Cell Biology Unit, Department of Immunology and Molecular Medicine, University of Sri Jayewardenepura, Nugegoda, Sri Lanka, 2 Colombo Municipal Council, Colombo, Sri Lanka, 3 MRC Human Immunology Unit, MRC Weatherall Institute of Molecular Medicine, University of Oxford, Oxford, United Kingdom

ʘ These authors contributed equally to this work.
* gathsaurie.malavige@ndm.ox.ac.uk

**Data Availability Statement:** All data is available in the manuscript, figures and supplementary data.

**Funding:** We are grateful to the funding by the Allergy Immunology and Cell Biology Unit of

## Abstract

### Background

There are limited data regarding the safety and immunogenicity of the Sinopharm/BBIBP-CorV vaccine in pregnancy. Therefore, we sought to investigate the antibody responses and maternal and fetal adverse events following this vaccine in pregnant mothers in Sri Lanka.

### Methods and findings

SARS-CoV-2 receptor binding domain (RBD) specific total antibodies and ACE2 blocking antibodies were measured by ELISA in pregnant mothers (n = 94) who received the vaccine in the first (n = 2), second (n = 57) and third (n = 33) trimester of pregnancy. Data regarding adverse events and fetal and maternal outcomes were obtained from the women once they delivered. No adverse maternal or fetal complications reported such as miscarriage, thrombotic events, hypertensive disorders, fetal death, preterm delivery, or congenital anomalies were reported. 58/94 (61.7%) had RBD binding antibodies and were found to be seropositive at the time of recruitment. All women seroconverted after the second dose and 31/36 previously uninfected women and 57/58 previously infected women gave a positive response to ACE2 blocking antibodies. The RBD binding antibody levels (p = 0.0002) and ACE2 blocking antibodies (p<0.0001) were significantly higher in previously infected individuals post-second dose compared to uninfected individuals.

### Conclusions

The Sinopharm/ BBIBP-CorV vaccine appeared safe and induced high seroconversion rates and ACE2 blocking antibodies in pregnant mothers in the second and third trimester in pregnancy. However, the RBD binding antibodies and ACE2 blocking antibodies post-second dose were significantly higher in previously infected pregnant mothers post-second

University of Sri Jayewardenepura (CJ, GNM), UK
Medical Research Council (GSO) and the Foreign
and Commonwealth Office (GNM) for support. The
funders had no role in study design, data collection
and analysis, decision to publish, or preparation of
the manuscript.

**Competing interests:** The authors have declared
that no competing interests exist.

dose, suggesting that two doses of the vaccine are likely to be less immunogenic in previously unexposed individuals.

## Introduction

SARS-CoV-2 infection in pregnancy is associated with a higher risk of maternal complications, severe illness and neonatal complications such as preterm delivery and low birth weight babies [1]. Critical care admissions, still births and early neonatal deaths were found to be significantly higher in unvaccinated pregnant women compared to those who were fully vaccinated [2]. Therefore, all countries recommend COVID-19 vaccines and boosters for all pregnant mothers in order to prevent maternal and neonatal complications [3, 4].

The mRNA vaccines BNT162b2 (Pfizer–BioNTech) and mRNA-1273 (Moderna) and AZD1222 (ChAdOx1 nCoV-19) were shown to be safe in pregnancy in all trimesters [5–7]. It was also shown that the BNT162b2 (Pfizer–BioNTech) induced robust antibody responses in pregnant mothers and high titres of SARS-CoV-2 spike protein specific antibodies in cord blood [8, 9]. Immunization of pregnant women with a mRNA vaccine was shown to reduce hospitalization of infants <6 months of age by 32% to 80%, depending on the trimester in which the vaccine was administered [10]. Although there are a few studies showing the immunogenicity and safety of mRNA and adenovirus vector vaccines (ChAdOx1 nCoV-19) in pregnancy, there are limited data on immunogenicity and safety of inactivated COVID-19 vaccines in pregnant mothers.

Sri Lanka experienced significantly morbidity and mortality due to COVID-19 outbreaks from April to October 2021 due the alpha and delta variants [11]. As a result, many pregnant mothers were also hospitalized with COVID-19, resulting in significant morbidity prior to administration of COVID-19 vaccines [12]. The Sinopharm/BBIBP-CorV vaccine was the main vaccine used in the primary vaccination series in Sri Lanka, and as of 24th of February 2022, 11 million individuals of the 21.9 million population had been fully immunized with this vaccine [13]. The Sinopharm/BBIBP-CorV was recommended for all pregnant mothers in Sri Lanka for their primary vaccination series. We previously showed that the Sinopharm/BBIBP-CorV induced high seroconversion rates, ACE2 blocking antibodies soon after administration of the second dose of the vaccine [14], although antibody responses declined in all age groups, especially in those >60 years, 12 weeks post-second dose [15]. As there are limited data regarding the safety and immunogenicity of the Sinopharm/BBIBP-CorV vaccine in pregnant mothers, we sought to investigate the antibody responses and maternal and fetal adverse events following this vaccine in pregnant mothers in Sri Lanka.

## Materials and methods

### Study participants

Ninety-four (94) pregnant mothers aged 18 years and above, who were registered with the maternal and child health clinics in the Colombo Municipal Council region of Sri Lanka were recruited following informed written consent. Those who were able to give a blood sample at the time of receiving the first and second doses and again 6 to 12 weeks after the second dose were included in the study. These individuals received the first dose of the vaccine between mid July 2021 to mid-August 2021. The first blood sample was obtained at the time of recruitment (the time they received the first dose of the vaccine) to determine past infection with the SARS-CoV-2 virus. The second blood sample was obtained at 4 weeks from the first blood

sample, when they received the second dose and the third blood sample was obtained 6 to 12 weeks from the time of recruitment (since obtaining the first blood sample). Data regarding adverse events and fetal and maternal outcomes were obtained from the women once they delivered.

Ethics approval was obtained from the Ethics Review Committee of the University of Sri Jayewardenepura.

### Detection of SARS-CoV-2 specific antibodies

The presence of total antibodies (IgM, IgG or IgA) to the receptor binding domain (RBD) of the SARS-CoV-2 virus was determined by using the Wantai SARS-CoV-2 total antibody ELISA (Beijing Wantai Biological Pharmacy Enterprise, China). Based on the manufacturer's instructions a cut-off value for each ELISA was calculated and according to this value, the antibody index (which is used as an indirect indicator of the antibody titre) was calculated by dividing the absorbance of each sample by the cut-off value. This assay was found to be 100% specific in the Sri Lankan population previously by evaluating this assay in serum samples collected prior to 2019 [16].

### Surrogate neutralizing antibody test (sVNT) to detect ACE2 receptor blocking antibodies

The surrogate virus neutralization test (sVNT) (Genscript Biotech, USA), which detects the presence antibodies that inhibit the binding of the RBD to the ACE2 was used to measure the presence of neutralizing antibodies in the vaccine recipients. These was carried out according to the manufacturer's instructions as previously described by us and the ACE2 blocking antibody levels expressed % of inhibition of binding [17]. An inhibition percentage $\geq 25\%$ in a sample was considered as positive for ACE2 blocking antibodies as previously described by us in the Sri Lankan population [17].

### Statistical analysis

GraphPad Prism version 8.3 was used for statistical analysis. The differences in antibody responses after the first and second doses of the vaccine was analyzed using the Wilcoxon matched-pairs signed rank test. The differences in the antibody titres between infected and uninfected pregnant mothers was determined by the Mann-Whitney test (two-tailed). Spearman's correlation coefficient was used to determine the correlation between antibody responses and the age of the individuals.

## Results

Of the ninety-four (94) pregnant mothers, 58 (61.7%) were between the ages of 18 to 30 years and 36 (38.3%) were between 31 to 45 years. 2/94 (2.1%) women obtained the vaccine in the first trimester, 57/94 (60.6%) in the second trimester and 33 (35.1%) in the third trimester. There were no adverse maternal or pregnancy related complications reported such as miscarriage, thrombotic events or hypertensive disorders. There were no adverse fetal outcomes such as fetal death, preterm delivery or congenital anomalies. A cardiac anomaly was seen in one baby, whose mother had gestational diabetes and it was not considered to be related to the vaccine.

## Antibody responses to the receptor binding domain

Fifty-eight out of the ninety-four (58/94–61.7%) had SARS-CoV-2 specific total antibodies at the time of recruitment (at the time of receiving the first dose) and therefore, we considered to be previously infected with the SARS-CoV-2 virus. 2/58 were diagnosed as being infected with the SARS-CoV-2 virus by a positive RT-qPCR, 4 to 8 weeks prior to administration of the first dose of the vaccine. The other women did not know that they had previously been infected with the virus. Of the 36 individuals who were found to be uninfected at baseline, 2 individuals developed COVID-19 and were tested positive 2 to 4 weeks since obtaining the second dose of the vaccine.

At 4 weeks after the 1st dose (at the time of receiving the second dose), 30/36 (83.3%) of those who were uninfected had seroconverted, and 2 to 8 weeks after the 2nd dose (6 to 12 weeks after the first dose), all 36 women had seroconverted. The total antibody index (an indirect measure of the antibody titres against antibodies that bind the receptor binding domain), was significantly higher from baseline to 4 weeks after the first dose in those who were uninfected ($p < 0.0001$) and infected ($p = 0.0003$) (Fig 1). However, there was no difference in the total antibody titres in uninfected individuals 4 weeks after the first dose and 6 to 12 weeks after the first dose (2 to 8 weeks after 2nd dose) in uninfected ($p = 0.09$) and infected ($p = 0.40$) women. However, the total antibody titre to the RBD was significantly higher at 4 weeks after the first dose ($p < 0.0001$) and 6 to 12 weeks after the 1st dose ($p = 0.0002$) in baseline infected individuals compared to uninfected women (Fig 1). There was no association with the age of the pregnant mothers with the total antibody titres following the second dose in those who were baseline infected (Spearman's r = -0.19, $p = 0.15$) or uninfected (Spearman's r = -0.18, $p = 0.29$).

We then proceeded to compare the total antibody responses in uninfected pregnant mothers who received two doses of the Sinopharm/BBIBP-CorV vaccine, with non-pregnant females. We used our previously published data of total antibody responses in non-pregnant women aged 20 to 45, to compare the seroconversion rates and antibody levels two weeks post second dose of the Sinopharm/BBIBP-CorV [14]. There was no difference in the seroconversion rates or the total antibody titres in pregnant mothers compared to non-pregnant women ($p = 0.39$).

## ACE2 blocking antibodies in infected and uninfected pregnant women

Sixteen of the fifty-eight (16/58–27.5%) women who were found to be seropositive at baseline (at the time of recruitment), did not have ACE2 blocking antibodies above the positive cut off threshold (>25% of inhibition). After the first dose, 3/16 individuals baseline infected individuals developed ACE2 blocking antibodies beyond the positive threshold and all developed antibodies beyond the positive threshold after both doses. 12/36 (33.3%) uninfected individuals gave a positive result for ACE2 blocking antibodies at 4 weeks and 31/36 (86.1%) individuals gave a positive result after both doses. 55/58 (94.8%) of baseline infected women gave a positive result after the first dose, while 57/58 (98.3%) individuals gave a positive response after the second dose.

ACE2 blocking antibodies significantly rose from baseline to 4 weeks ($p < 0.0001$) and then 6 to 12 weeks ($p = 0.003$) after the first dose in uninfected individuals (Fig 2). The ACE2 blocking antibodies also significantly increased in those who were baseline infected at 4 weeks from the first dose ($p < 0.0001$) and 6 to 12 weeks ($p = 0.02$) after the second dose. Baseline infected individuals had significantly higher ACE2 blocking antibodies at 4 weeks from the first dose ($p < 0.0001$) and 6 to 12 weeks from the first dose ($p < 0.0001$) compared to baseline uninfected individuals (Fig 2). There was no association with age and the ACE2 antibody levels in baseline

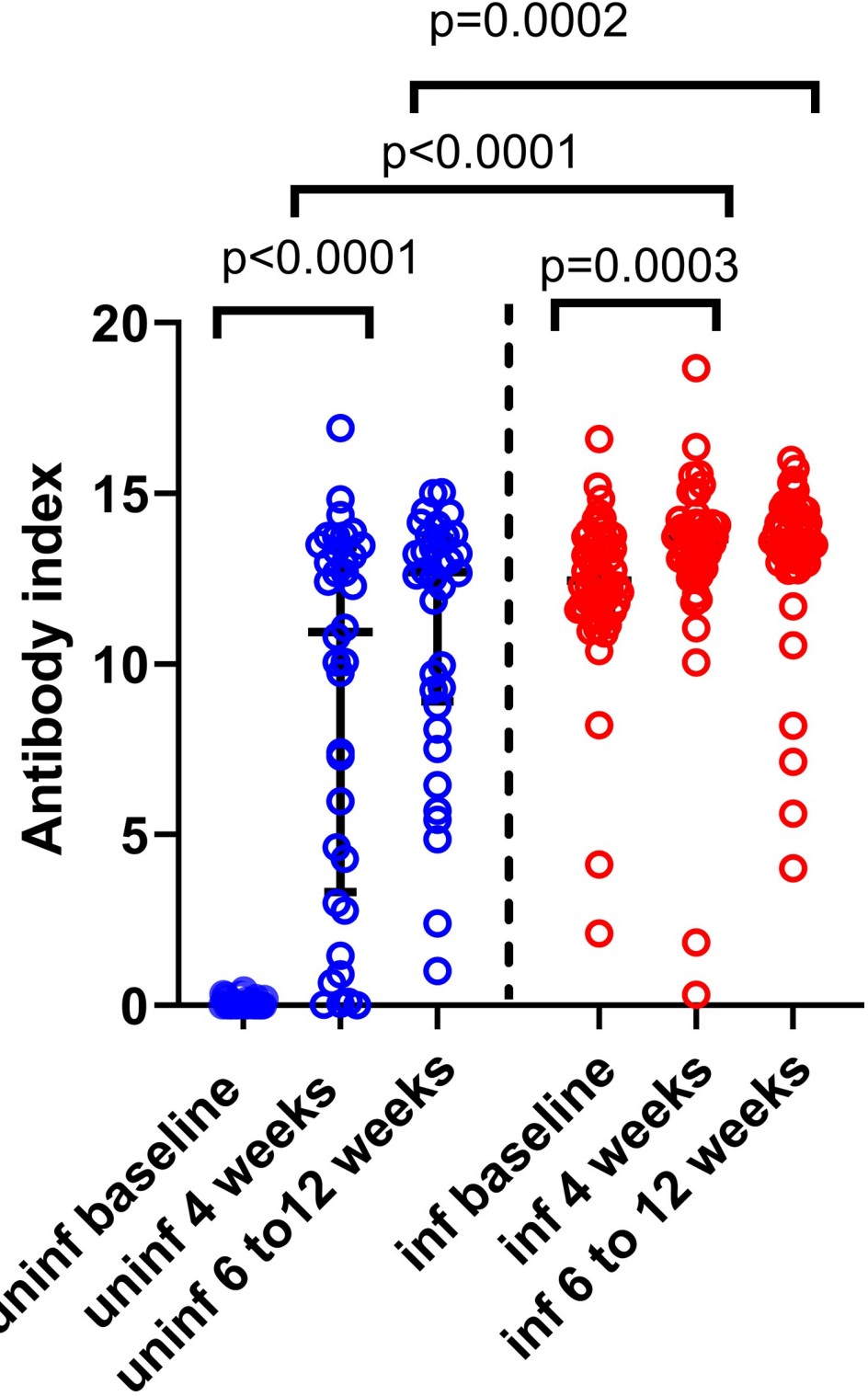

**Fig 1. Antibody responses to the receptor binding domain of SARS-CoV-2 virus, following first and second dose of the Sinopharm/ BBIBP-CorV vaccine.** SARS-CoV-2 specific total antibodies were measured in baseline uninfected (n = 36) and infected (n = 58) pregnant women by ELISA, at baseline, 4 weeks from the first dose and 6 to 12 weeks following the second dose (2 to 8 weeks following the second dose). Wilcoxon matched-pairs signed rank test was used to compare the means of the antibody indexes, before and after the vaccine, while the Mann-Whitney U test was used to calculate the differences in antibody responses at each time point in baseline infected and uninfected women. All tests were two-tailed. The lines indicate the median and the inter quartile range.

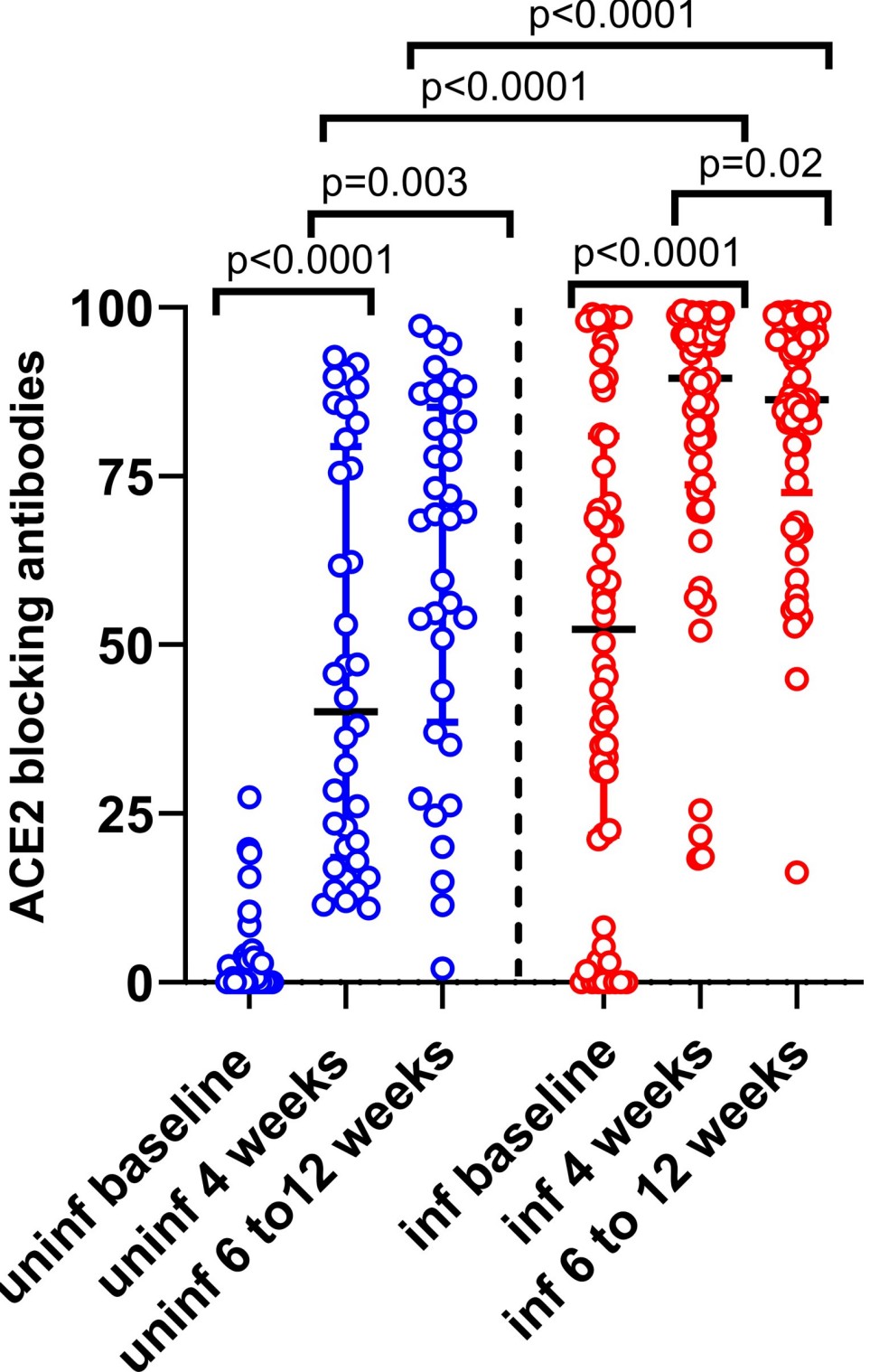

**Fig 2. ACE2 blocking antibodies following first and second dose of the sinopharm/ BBIBP-CorV vaccine.** The ACE2 receptor blocking antibodies were measured by the surrogate virus neutralizing test in baseline uninfected (n = 36) and infected (n = 58) pregnant women. Wilcoxon matched-pairs signed rank test was used to compare the means of the antibody indexes, before and after the vaccine, while the Mann-Whitney U test was used to calculate the differences in antibody responses at each time point in baseline infected and uninfected women. All tests were two-tailed. The lines indicate the median and the inter quartile range.

uninfected (Spearman's r = -0.13, p = 0.43) and infected individuals (Spearman's r = 0.19, p = 0.16).

## Discussion

In this study we assessed the adverse events, safety and immunogenicity of the Sinopharm/ BBIBP-CorV vaccine in pregnant mothers in Sri Lanka. We found that the vaccine appeared safe and did not cause any maternal and fetal adverse outcomes in the second and third trimesters. The only adverse fetal outcome, which was the presence of a cardiac anomaly was seen in a neonate of a mother with gestational diabetes mellitus and was not considered to be related to the vaccine. All baseline infected and uninfected women seroconverted after two doses of the vaccine. ACE2 blocking antibodies above the positive cut off threshold was detected in 86.1% of baseline uninfected women and 98.3% of the baseline infected individuals after two doses of the vaccine. These seroconversion rates and ACE2 antibody detection rates were similar to rates reported in non-pregnant individuals of the same age, as previously reported [14].

The antibody levels to the RBD of SARS-CoV-2 virus were significantly higher in baseline infected individuals post-first dose and post-second dose compared to baseline uninfected individuals. Interestingly, the RBD binding antibody levels did not significantly increase following the second dose, compared to levels in the first dose in either baseline uninfected or infected women. However, a significant increase was seen with the ACE2 blocking antibodies in both baseline uninfected and infected women following the second dose compared to the first. As ACE2 blocking antibodies measured by the sVNT assay was shown to be a surrogate marker of neutralizing antibodies (Nabs) [18], the vaccine appears be important to induce higher levels of Nabs. However, both RBD binding antibodies and ACE2 blocking antibodies were significantly higher post-first dose and post second-dose following the Sinopharm/ BBIBP-CorV vaccine, in previously infected individuals compared to uninfected. It has been shown that significantly higher RBD binding antibodies and neutralizing antibodies were detected in both SARS-CoV-2 infected and uninfected individuals following the second dose of the two mRNA vaccines (BNT162b2 and mRNA-1273) and AZD1222 [19]. However, the antibody levels did not increase significantly in those with previous infection after post-second dose from levels seen post-first dose in previously infected individuals [20]. In fact, the antibody levels after one or two doses of a BNT162b2 was found to be similar in previously infected individuals [21]. In our cohort, the antibodies to the RBD and ACE2 blocking antibodies were significantly increased post-second dose, compared to post-first dose in both uninfected and infected individuals. This could be because as the inactivated vaccines have shown to be less immunogenic than the mRNA vaccines, as previously reported [22].

Although 61.7% of women were found to be baseline infected (presence of SARS-CoV-2 RBD binding antibodies), 27.5% did not have ACE2 blocking antibodies above the positive threshold. Since only 2/58 women had a symptomatic/confirmed SARS-CoV-2 infection, it is possible that they had an asymptomatic or mild illness. It has been shown that ACE2 blocking antibodies were not detectable in some individuals who had mild or asymptomatic illness, while in many individuals who developed mild illness, they declined with time [17, 23].

A large proportion of women (61.7%) were found to be seropositive for the SARS-CoV-2 virus at the time of recruitment (when they received the first dose of the vaccine), which shows that by mid- July 2021, many individuals in the community in the Colombo Municipality Council region were naturally infected. We previously showed that by end of January 2021, the overall seroprevalence rates of this population was 24.5% [18]. The number of cases markedly increased from mid-April to June due to the alpha variant [19], while the outbreak due to the delta variant was predominantly from mid-July 2021 to November 2021 [24]. Therefore, these

individuals were recruited to the study, while the cases due to the delta variant was starting rise. Although, the seropositivity rates in this small cohort of pregnant mothers are unlikely to reflect the seroprevalence rates in the community in Colombo, it does suggest that prior to the large outbreak due to delta occurred, there was likely to have been large number of individuals in the community, who were naturally infected.

## Conclusions

We have shown that the Sinopharm/ BBIBP-CorV vaccine appeared safe and induced high seroconversion rates and ACE2 blocking antibodies in pregnant mothers in the second and third trimesters in pregnancy. However, the RBD binding antibodies and ACE2 blocking antibodies post-second dose was significantly higher in previously infected pregnant mothers post-second dose, suggesting that two doses of the vaccine is likely to be less immunogenic in previously unexposed individuals.

## Supporting information

**S1 Data. De-identified primary data that was used in the analysis and for generation of figures.**
(XLSX)

## Author Contributions

**Conceptualization:** Chandima Jeewandara, Gathsaurie Neelika Malavige.

**Data curation:** K. A. Chintha S. Jayampathi, Inoka Sepali Aberathna, Osanda Dissanayake, Nayanathara Gamalath, Dinithi Ekanayake, Jewantha Jayamali, Gayasha Somathilake.

**Formal analysis:** Gathsaurie Neelika Malavige.

**Funding acquisition:** Chandima Jeewandara, Graham S. Ogg, Gathsaurie Neelika Malavige.

**Investigation:** Thushali Ranasinghe, Banuri Gunasekara.

**Methodology:** Thushali Ranasinghe, Inoka Sepali Aberathna, Saubhagya Danasekara, Thashmi Nimasha, Heshan Kuruppu, Achala Kamaladasa.

**Project administration:** Chandima Jeewandara, K. A. Chintha S. Jayampathi, Dinuka Guruge, Achala Kamaladasa.

**Resources:** Chandima Jeewandara, K. A. Chintha S. Jayampathi, Ruwan Wijayamuni.

**Supervision:** Chandima Jeewandara, K. A. Chintha S. Jayampathi, Dinuka Guruge, Ruwan Wijayamuni.

**Writing – original draft:** Thushali Ranasinghe, Gathsaurie Neelika Malavige.

**Writing – review & editing:** Graham S. Ogg, Gathsaurie Neelika Malavige.

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
