## [Decision Letter · Decision Letter 0]

9 May 2022

PGPH-D-22-00328

Antibody responses to Sinopharm/BBIBP-CorV in pregnant individuals in Sri Lanka

Dear Dr. Malavige,

Thank you for submitting your manuscript to PLOS Global Public Health. After careful consideration, we feel that it has merit but does not fully meet PLOS Global Public Health’s publication criteria as it currently stands. Therefore, we invite you to submit a revised version of the manuscript that addresses the points raised during the review process.

The reviewers had only minor comments. Yet please address them fully and in your response to reviewers document, write out what you changed in your manuscript and where you made those changes.

Please submit your revised manuscript by . If you will need more time than this to complete your revisions, please reply to this message or contact the journal office at globalpubhealth@plos.org. Please include the following items when submitting your revised manuscript:

We look forward to receiving your revised manuscript.

Kind regards,

Abram Luther Wagner, PhD, MPH

Academic Editor

Journal Requirements:

1. If you have no competing interests to declare, please state "The authors have declared that no competing interests exist", please update the Competing Interest section in the system.

- State the initials, alongside each funding source, of each author to receive each grant.

.”

Additional Editor Comments (if provided):

Reviewers' comments:

Reviewer's Responses to Questions

**Comments to the Author**

1. Does this manuscript meet PLOS Global Public Health’s publication criteria? Is the manuscript technically sound, and do the data support the conclusions? The manuscript must describe methodologically and ethically rigorous research with conclusions that are appropriately drawn based on the data presented.

Reviewer #1: Partly

Reviewer #2: Yes

2. Has the statistical analysis been performed appropriately and rigorously?

Reviewer #1: Yes

Reviewer #2: Yes

3. Have the authors made all data underlying the findings in their manuscript fully available (please refer to the Data Availability Statement at the start of the manuscript PDF file)?

Reviewer #1: No

Reviewer #2: No

4. Is the manuscript presented in an intelligible fashion and written in standard English?

Reviewer #1: Yes

Reviewer #2: Yes

5. Review Comments to the Author

Reviewer #1: This study explores a timely topic and can provide relevant information on COVID-19 response.

Data availability: PLOS journals require authors to make all data necessary to replicate their study’s findings publicly available without restriction at the time of publication. When specific legal or ethical restrictions prohibit public sharing of a data set, authors must indicate how others may obtain access to the data.

Regarding sampling, what was the inclusion and exclusion criteria for participants and justifications (if any) ? why is the sample size low?

Was the distribution of participants among the 3 trimesters by design or co-incidental, how did this affect the study outcome?

Reviewer #2: The authors use “Pregnant individuals”, “Pregnant Women”, and “pregnant females” should be consistent with the population. I would suggest they use “Pregnant mothers”

Materials and methods

Start line 107 with a word not a number (i.e. Ninety-four (94) females above….) and line 167

Was it part of the inclusion criteria for all the participants to be above 18 years? Or it was 18 years and above. Clarify on this.

“These individuals received the first dose of the vaccine between mid July 2021 to

110 mid-August 2021” Were all the participants in the same trimester of their pregnancy? Add more information on this.

6. PLOS authors have the option to publish the peer review history of their article (what does this mean?). If published, this will include your full peer review and any attached files.

**Do you want your identity to be public for this peer review?** For information about this choice, including consent withdrawal, please see our Privacy Policy.

Reviewer #1: **Yes: **Dr. Samuel Kidane

Reviewer #2: No

---

## [Editor Report · Decision Letter 1]

19 May 2022

Antibody responses to Sinopharm/BBIBP-CorV in pregnant mothers in Sri Lanka

PGPH-D-22-00328R1

Dear Professor Malavige,

We are pleased to inform you that your manuscript 'Antibody responses to Sinopharm/BBIBP-CorV in pregnant mothers in Sri Lanka' has been provisionally accepted for publication in PLOS Global Public Health.

Best regards,

Abram Luther Wagner, PhD, MPH

Academic Editor

Thank you for your contribution and your comprehensive response to the reviewers.